

# Characterisation of false-positive observations in botanical surveys

Quentin J. Groom[1,*] and Sarah J. Whild[2,*]

[1] Botanic Garden Meise, Meise, Belgium
[2] School of Science and the Environment, The Manchester Metropolitan University, Manchester, United Kingdom
[*] These authors contributed equally to this work.

## ABSTRACT

Errors in botanical surveying are a common problem. The presence of a species is easily overlooked, leading to false-absences; while misidentifications and other mistakes lead to false-positive observations. While it is common knowledge that these errors occur, there are few data that can be used to quantify and describe these errors. Here we characterise false-positive errors for a controlled set of surveys conducted as part of a field identification test of botanical skill. Surveys were conducted at sites with a verified list of vascular plant species. The candidates were asked to list all the species they could identify in a defined botanically rich area. They were told beforehand that their final score would be the sum of the correct species they listed, but false-positive errors counted against their overall grade. The number of errors varied considerably between people, some people create a high proportion of false-positive errors, but these are scattered across all skill levels. Therefore, a person's ability to correctly identify a large number of species is not a safeguard against the generation of false-positive errors. There was no phylogenetic pattern to falsely observed species; however, rare species are more likely to be false-positive as are species from species rich genera. Raising the threshold for the acceptance of an observation reduced false-positive observations dramatically, but at the expense of more false negative errors. False-positive errors are higher in field surveying of plants than many people may appreciate. Greater stringency is required before accepting species as present at a site, particularly for rare species. Combining multiple surveys resolves the problem, but requires a considerable increase in effort to achieve the same sensitivity as a single survey. Therefore, other methods should be used to raise the threshold for the acceptance of a species. For example, digital data input systems that can verify, feedback and inform the user are likely to reduce false-positive errors significantly.

Corresponding author
Quentin J. Groom,
qgroom@reticule.co.uk,
quentingroom&gmail.com

## INTRODUCTION

Errors in science are inevitable. Sometimes they are the result of random chance, but also from human fallibility. Errors are particularly common in observations of biodiversity, where organisms can be either inconspicuous, hard to identify or hidden. Furthermore, the process of observation can be disrupted by external influences and observer biases (*Simons et al., 2007*; *Willson, Winne & Todd, 2011*). Animals are often intentionally secretive, but

even sedentary organisms, such as plants, are difficult to observe owing to their similarity to each other. These sorts of errors lead to false-negative errors. False-negative errors are expected in plant surveys due to the variability in the detectability of different species in different seasons and habitats (*Rich & Woodruff, 1992*; *Chen et al., 2009*; *Chen et al., 2013*; *Morrison, 2016*). False-positive errors however, are those arising from observing something that was not there. These errors have been given many names, including detection errors, type 1 errors, errors of commission and misclassifications. Here we have chosen to use the terms false-positive and false-negative for the sake of readability. False-positive errors occur for several reasons; observers can either misidentify an organism; wrongly report the date or location or incorrectly transcribe otherwise correctly reported data.

While errors in vegetation surveying have been studied from many aspects there are few studies on false-positive errors in botanical recording and few survey schemes have specific quality assurance mechanisms, such as suggested by *Scott & Hallam (2003)*. In contrast, much more attention has been paid to observation errors of animals where progress has been made in the methods for observation and analysis (*Royle & Link, 2006*; *Elphick, 2008*). Much expert plant identification of common taxa in the field is done using gestalt perception, rather than formal identification of characters (*Ellis, 2011*). However, human senses and reasoning, though remarkable, are prone to various sorts of error including apophenia (seeing apparently meaningful patterns within random data), generalizations and confirmation bias.

Users of botanical records expend considerable effort in "cleaning" data (*Chapman, 2005*). Cleaning entails using experience to verify and reformat the results of biological surveys, however, this is an inefficient process as errors are much better resolved close to their sources. Furthermore, data "cleaning" is also fallible, and elimination of errors early in the workflow is likely to be quicker and less costly than when time and distance is put between the observation event and the person analysing it. Statistical methods can also be used to account for observer errors and bias (*Royle & Link, 2006*; *Miller et al., 2011*; *Bird et al., 2014*; *Dorazio, 2014*), but while these approaches are useful, the first line of defence should be minimizing errors in field surveying.

In biological surveying, false-positive observations can be more costly than false-negatives. If false-negatives are suspected, additional surveying can help to resolve them. Also false-negatives are more likely to occur for rare species thus the more surveying conducted, the more one becomes confident that the species is either truly absent or at least extremely rare within your survey area. In the case of threatened species, false-negatives could lead to inappropriate actions in planning decisions or site management, but false-positives give the impression that a species is more common than it is and may lead to its conservation status not being recognised. The costs of errors needs to be assessed based on the goals of the survey, and should be considered in survey design and analysis methods (*Fielding & Bell, 1997*).

In contrast to false-negatives, false-positive errors are difficult to refute and can pollute datasets indefinitely. One only has to think of the time and effort wasted on extreme cases of false-positive errors, such as observations of plesiosaurs in Loch Ness and hominids in the Rocky Mountains, but there are many other examples (*Sabbagh, 2001*;
*McKelvey, Aubry & Schwartz, 2008*). Even in more benign cases *Royle & Link (2006)* showed, using a simulation, that ignoring false-positive observations in occupancy models could result in large biases in occupancy estimates (see also *Molinari-Jobin et al., 2012*).

In this study we use a quite unique set of plant surveying data where the same sites have been surveyed repeatedly by many independent observers. These data are derived from tests for Field Identification Skills Certificates (FISCs) that have been running for eight years under the aegis of the Botanical Society of Britain and Ireland. These certificates are intended to give the participants and potentially their future or current employers a guide to their skill at vascular plant identification in the field. These day-long tests include two laboratory-based tests and an afternoon field test and it is from this field test that the data in this paper are derived.

In classical signal detection theory, where the signal has to be separated from the noise, we can reduce the number of false-positive errors by increasing the detection threshold for accepting the signal. However, this is at the expense of an increase in false-negative errors (*Wolf et al., 2013*). We can examine this trade-off by changing the acceptance criteria for a species to be present. One simple method to increase the detection threshold is to combine the results of two or more observers, only accepting observations if multiple observers agree. So-called double-observer methods are frequently used in animal surveys, particularly for avian point counts (*Simons et al., 2007*; *Conn et al., 2013*).

Double-observer methods can reduce the false-positive observations because false-positives are rare. If false-positives always account for a small proportion of the total number of observations and an observer picks their false-positive observations randomly from a fairly large pool of species names, then the number of false-positive observations of any one species should always be small and the chances of two observers picking the same false-positive species is extremely small.

However, there are two potential problems with this approach. Firstly, all the species that are actually present but only observed by one observer become false-negative observations. Secondly, the assumption that observers are unbiased in their creation of false-positive observations may not be true. For example, observers may pick false-positive observations from species with similar character, such as those they are phylogenetically closely related to or they may pick them from common plants that they assume to be present.

In this paper we examine the characteristics of false-positive observations from the perspective of plant detectability, phylogenetic relatedness and their familiarity to observers. We examine whether changing the threshold for a true positive observation improves the accuracy of surveys and we discuss what other strategies could be used to reduce errors in botanical surveying. The intention is that the results can be used to design better plant survey methods that will lead to a reduction in the number of false-positive observations.

## MATERIALS & METHODS

### Sites descriptions

This analysis is based on the field test data collected from 238 surveys from the FISCs conducted in Shropshire, UK. From 2007 to 2014 six sites have been used: Sweeney

**Table 1** **A summary of the survey sites.** The dates of the surveys and total numbers of surveys conducted, their location and habitat.

| Site name | Survey dates | Number of surveys | Location (WGS84) | Habitat |
|---|---|---|---|---|
| Windmill Hill | 27/06/2009 16/07/2009 01/07/2010 | 53 | N52°36′13″ W2°33′18″ | Grassland |
| Ballstone Quarry | 04/07/2012 11/07/2012 | 35 | N52°35′35″ W2°34′35″ | Grassland |
| Blakeway Hollow | 20/07/2011 31/07/2011 | 38 | N52°35′37″ W2°34′57″ | Grass verges and also dense species-rich hedges |
| Sweeney Fen | 06/08/2007 | 20 | N52°49′4″ W3°4′38″ | Neutral grassland and calcareous fen |
| Old River Bed | 24/06/2008 28/06/2008 14/09/2008 | 49 | N52°43′40″ W2°44′47″ | Sedge swamp |
| Aston Locks | 02/07/2014 09/07/2014 | 42 | N52°49′51″ W2°59′18″ | Areas of open water, neutral grassland, tall herb and some sedge swamp |

Fen; Ballstone Quarry; Windmill Hill; Aston Locks off-line reserve; The Old River Bed, Shrewsbury and Blakeway Hollow. A summary of the sites and the surveys conducted on them are presented in Table 1. The sites were chosen to fit the following criteria, which are consistent with the FISC protocols. They are around 2–3 hectares of accessible habitat, which are relatively safe in health and safety terms. They were selected for their species rich habitat, such as unimproved grassland with some scrub areas, or short fen, or not too wet sedge swamp, or broad-leaved woodland. The area to be surveyed was made very clear to the participants—if a smaller area was used within a larger reserve it was fenced or taped off clearly. It was made clear to participants whether or not hedges were to be surveyed.

The site selection criteria were as follows.

- A relatively small more or less homogeneous site with fairly distinct boundaries.
- Small enough to survey thoroughly within two hours.
- Large enough for individual surveys to be carried out and for invigilation to be effective.
- Possessing a reasonably complex vegetation with a good range of grasses, sedges and/or rushes, giving a total of around 100 vascular plant species to record.

Windmill Hill and Ballstone Quarry are grassland sites on Silurian Limestone. Blakeway Hollow is also on Silurian Limestone but is a sunken trackway with grass verges and also dense species-rich hedges. Sweeney Fen is a mixture of neutral grassland and calcareous fen over Carboniferous Limestone. The Old River Bed is an old meander filled with sedge swamp on Quaternary deposits. Aston Locks off-line reserve is adjacent to the Montgomery Canal and has areas of open water, neutral grassland, tall herbs and some sedge swamp.

All sites were relatively easy to access over stiles or through gates. The Old River Bed is arguably the most challenging site as it can be wet, and Windmill Hill is on a steep slope but the sites were chosen to be as accessible as possible.

Demographic data on observers were not collected, but anecdotally the age range is between 25 and 60 with a median in the 30 s. The gender balance is roughly equal. The main motivation for participants for taking a FISC has been career enhancement, with ecological consultants forming the bulk of participants. Occasionally, participants repeated the FISC, after gaining some experience in the field, however they were never tested on the same site twice.

The FISC field tests were evaluated by the number of correct species recorded in the field, and a score based on false-positive errors. Observers were clearly informed before the test that they would lose marks for false-positive observations giving them a clear disincentive to make mistakes. If they were not able to identify a plant to species they could report a genus for which they received half marks, but they could only do this once for a genus.

FISC participants recorded against a 'gold-standard' observer—a volunteer at skill level 5 (level 5 is a professional level plant recorder who is competent to record in most habitats and areas in the UK) who recorded under the same conditions for the same length of time and scores for participants were calculated as a percentage of the 'gold-standard' observer's total.

## Data

The digitisation of the field data was carried out by a small team with instructions to pick out four categories of botanical records—correct species; unreasonable or known incorrect; 'mythical' species and 'cautious' records, with just a generic name. Correct species were those identified by the gold-standard observer, or species that had been recorded at the site by reliable recorders in other surveys of the site. Unreasonable species are those that have never been found at the site before. Mythical species were those that do not exist or do not occur in the UK. Data were simplified to species with taxonomy following *Stace (2010)*, except for the phylogenetic signal tests where the taxonomy was aligned with the Daphne phylogeny (*Durka & Michalski, 2012*). The number of species for each genus was taken from *Stace (2010)*, where species for each genus in the UK and Ireland are numbered.

Detection probabilities are calculated from the number of detections divided by the number of surveys.

The data used in this paper has been openly deposited in the Zenodo repository under http://doi.org/10.5281/zenodo.46662.

## Bootstrapping scripts

The acceptance threshold for accepting a true-presence observation was varied by combining the results of two or more surveys. If the species was observed in each of the combined surveys then is was accepted as present. Therefore, the more surveys that are combined, the higher the threshold is to accept the presence of the species. Obviously, different surveys have varying lists of observed species so to measure the average numbers of false-positive and false-negative observations when surveys were combined a bootstrapping approach was used. Depending on the number of surveys to be tested, a random selection of surveys was selected from the pool of all surveys conducted at each site. The number of false-positives and true positives were calculated from this selection and average values were calculated from 10,000 randomly chosen selections. This script was written in Perl

(ActivePerl, version 5.16.3.1603) and is available in a public Github repository together with a sample input file (https://github.com/qgroom/grouped-surveys).

## Statistics

Statistics were performed in R version 3.1.0. Owing to the differences in species composition, species abundance and habit at each site, all analyses were conducted on each site separately and treated as individual replicates. Generalized linear models using the total number of false-positive observations resulted in overdispersed models, because so many of the species have either zero or one false-positive observation. The solution was to model the false-positives as a binary response variable, with a species either being a true-positive (zero) or a false-positive (one) in a survey. The proportion of false-positive observations was modelled using a generalized linear model using a logit link function. The mean interpolated 4 km$^2$ occupancy probability from southern England between 1995 and 2011 was used as a measure of the regional frequency of the species (*Groom, 2013*).

There are several tests for phylogenetic signal strength, particularly for continuous traits. However, as the majority of species had either one or zero false-positive observations, we again chose to treat false-positives as a binary trait. To test for a phylogenetic component to the false detection of species the D statistic was calculated following *Fritz & Purvis (2010)*. The D statistic is a measure of the phylogenetic signal strength of binary traits. The D statistic was calculated using the Caper package version 0.5.2 (*Orme, 2012*). The phylogeny used for the calculation was the Daphne phylogeny of Northern European plants (*Durka & Michalski, 2012*). To calculate the D statistic for generic observations, the Daphne tree was pruned using the Phytools package (*Revell, 2012*).

The specificity, in the context of botanical surveying, is the probability that a species is not seen given the plant is not present, it decreases with increasing numbers of false-positive observations and increases with the number of true-negatives (*Fielding & Bell, 1997*). The sensitivity is the probability that the plant is observed given that the plant is present, it decreases with increasing numbers of false-negative errors and increases with the number of true-positives. Mean specificity and sensitivity were calculated from the mean bootstrapped values for true positive, false-negative and false-positive observations. The value for true negative observations is somewhat arbitrary, because any number of species may not be present. Therefore, the number of true-negatives were calculated from the cumulative number of false-positive species recorded at the site, minus the average number of false-positive species in that survey or combination of surveys. This seems a reasonable value to use because the list of false-positive species are species of the British flora that were considered by the observers as being potentially present. Although this value is arbitrary, using other values only affects the absolute value of the specificity, not the relative values. Sensitivity is calculated from the number of true positive observations, divided by the sum of the true positive observations and the false-negatives. Specificity is calculated from the number of true negatives, divided by the sum of the true negatives and false-positives.

All confidence intervals quoted in the text are calculated using the t distribution.

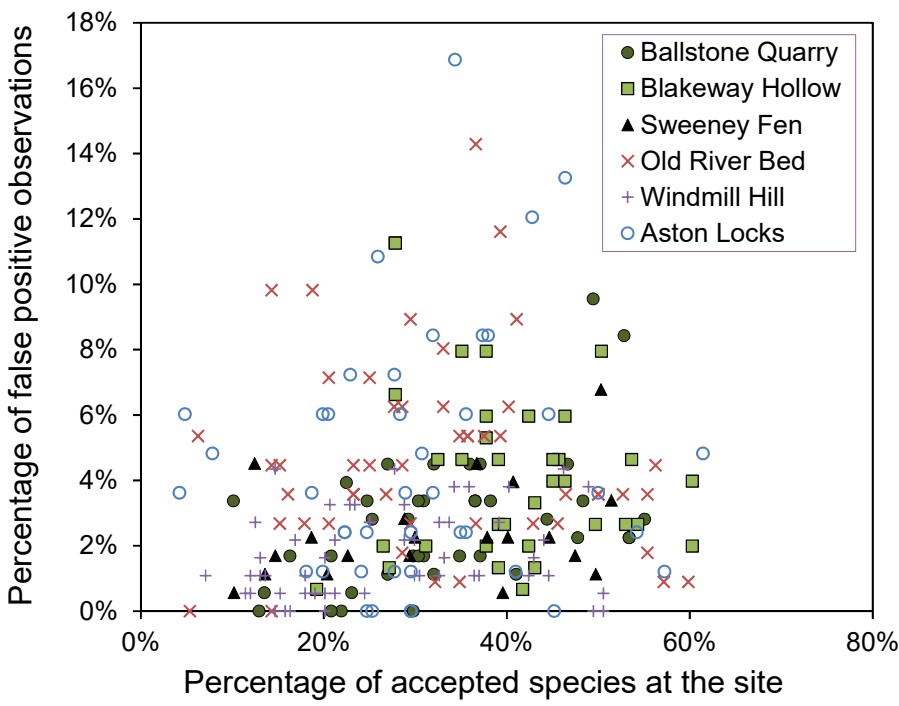

**Figure 1** **The percentage of false-positive verses correct observations for each observer.** The percentage of false-positive observations plotted against the percentage of correct observations from the list of accepted species at the site created by 'gold-standard' observers and other reliable surveys.

## RESULTS

The number of false-positive errors is not strongly related to the number of correct observations (Fig. 1). Across all sites the number of false-positive errors is not significantly correlated with the number of correct observations ($R = 0.14$ (95% CI [$-0.11$–$0.40$], $n = 6$)). Observers, who can identify many species, are not necessarily likely to create few false-positive errors. Some observers are cautious while others are reckless.

Figure 2 demonstrates that false-positives have a much lower detection probability than true positives. Based on the raw data, observers generate an average of 3.4% false-positive observations per survey. However, we have to accept that we could have made errors in our assessment of what species are at each site, which could either increase or decrease the number of apparent false-positive errors for the observers.

The detection probabilities of species accepted to be at the site vary widely, from close to one to close to zero (Fig. 2), though about 75% of species have a detection probability less than 0.5. The probability of a false positive observation occurring is always less than 0.5 and the majority of false detection probabilities are less than 0.05. Again, we should consider that the reference survey may have contained some false-negative errors, resulting in apparent false-positive errors for the observers. In such cases, it is likely that several of the observers will have observed these species and this may explain the right-hand tail of higher false detection probabilities seen in Fig. 2.

To investigate whether observers were more likely to make mistakes if there were many species to choose from in a genus we correlated the total number of false-positive

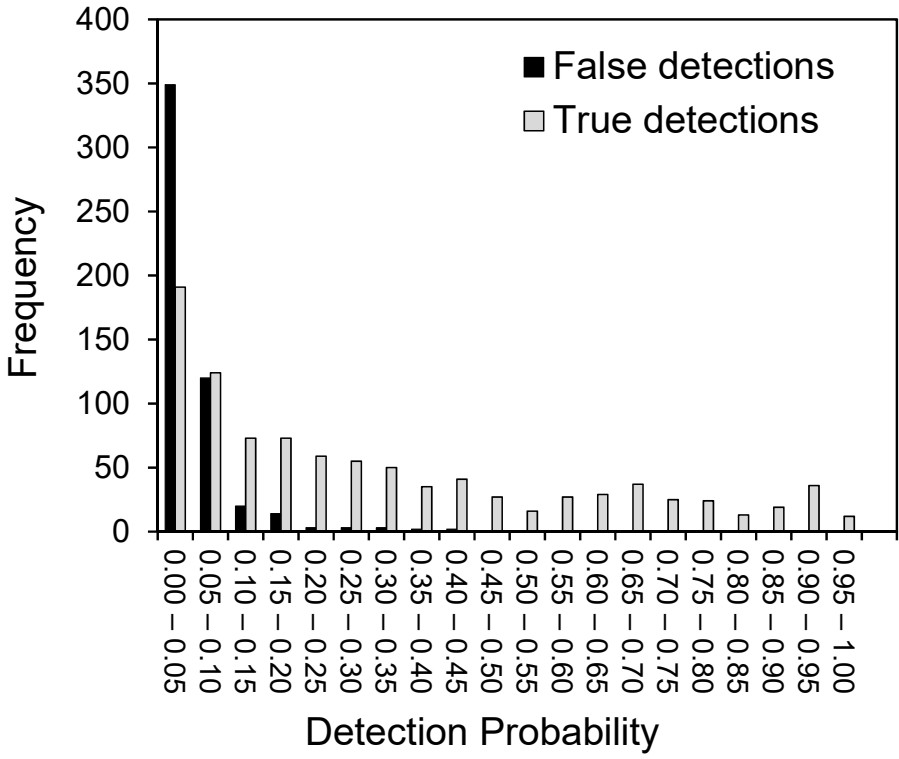

**Figure 2** **The variation of detection probability between different plant species.** The distribution of detection probabilities for true observations and false-positive observations. Detection probability is calculated from the number of detections divided by the number of surveys.

observations for members of a genus and the number of species in a genus in the UK. At all six sites this gave a significant positive correlation ($R = 0.34$ (95% CI [0.10–0.59], $n = 6$)).

## Is there a phylogenetic component to the false detection of species?

Values of D usually vary between 0 and 1 and are inversely proportional to the degree of phylogenetic clustering. Values less than zero are expected if phylogenetically closely related species are more often identified as false-positive errors. Values greater than zero occur where there is overdispersion, such that closely related species show opposite results. This would be the case where observers create errors that are closely related to the correct species. The D statistic is compared to the results from two evolutionary models, random and Brownian. Values close to 1 indicate that the random model is most appropriate and there is no phylogenetic relationship. The D statistic for false-positive observations at all six sites was close to one, indicating that there is no phylogenetic signal in the false detection of species (mean 1.04 (95% CI [0.98–1.11]), $n = 6$) (Table 2).

## Does familiarity with a species influence the false detection of species?

We use whether a species was chosen as a false-positive or a true positive as a binary independent variable and compared it with the average 4 km$^2$ occupancy for southern England. This allows us to compare how commonness and rarity relate to the likelihood

**Table 2  The phylogenetic signal strength of false-positive observations.** The $D$ statistic for the species observed at each site, showing the phylogenetic signal strength of false-positive observations. The $p$ value is the probability that the $D$ value fits the model. Numbers in square brackets are the numbers of true-positive and false-positive observations in the sample.

|  | Ballstone Quarry [173,86] | Blakeway Hollow [149,93] | Sweeney Fen [170,53] | Old River Bed [109,96] | Windmill Hill [177,71] | Aston Locks [162,91] |
|---|---|---|---|---|---|---|
| $D$ Statistic | 1.107 | 1.064 | 0.916 | 1.088 | 0.992 | 1.112 |
| $p$ Random model | 0.927 | 0.797 | 0.175 | 0.879 | 0.447 | 0.944 |
| $p$ Brownian model | < 0.01 | < 0.01 | < 0.01 | < 0.01 | < 0.01 | < 0.01 |
| $n$ | 259 | 242 | 223 | 205 | 248 | 253 |

of mis-observation in our surveys. For each site rarer species are more likely to be chosen in false positive observations (Fig. 3). From the y intercept of these models we can also estimate the probability of observing species that do not occur in southern England, which is 0.237 (95% CI [0.184–0.291], $n = 6$); though this seems a large proportion, this value is cumulative for all surveys conducted at the site.

## Inseparable taxa and fictional taxa

If observers were not able to identify a plant to species they could alternatively report a genus. Unlike errors these are conscious decisions to record a taxon at a less resolved taxon rank. The ten most recorded genera in this category were *Rosa*, *Salix*, *Euphrasia*, *Viola*, *Carex*, *Equisetum*, *Hypericum*, *Quercus*, *Epilobium* and *Myosotis*; listed in decreasing order of their number of observations. For all six sites there was a weak, but significant, positive correlation between the number of cautious observations and the number of species in a genus ($R = 0.19$ (95% CI [0.10–0.27], $n = 6$)). We did wonder whether observers who recorded more cautious taxa would record fewer false-positives. However, there is no evidence for this in these data. The number of cautious taxa recorded by an observer is not significantly correlated with their number of false-positive observations ($R = 0.06$ (95% CI [−0.19–0.07], $n = 6$)).

No significant phylogenetic pattern was seen in the recording of genera. The D statistic was calculated for all sites separately. The detailed results are not shown but the average D across sites was 0.85 (95% CI [0.62–1.07]) $n = 6$.

In addition to cautious identifications there were non-existent taxa. These were a mixture of imaginary Latin and English names. They only made up a small number of observations, on average 3.7 observations per site. None were completely imaginary and most were combinations of words that exist in other names, but not together. For example, there were incorrect Latin combinations such as *Calystegia arvense*, *Carex articulatus*, *Geranium palustre*, *Plantago ovalifolium* and *Polygonum vulgare*. Imagined vernacular names included black alkanet, burweed, separated rush and sheep's-beard. Names such as burweed do exist for species that do not occur in the British Isles, but it is assumed that the species intended by the observer was one that occurs in the British Isles with burred fruits, such as *Galium aparine*, *Geum urbanum* and *Torilis japonica*.

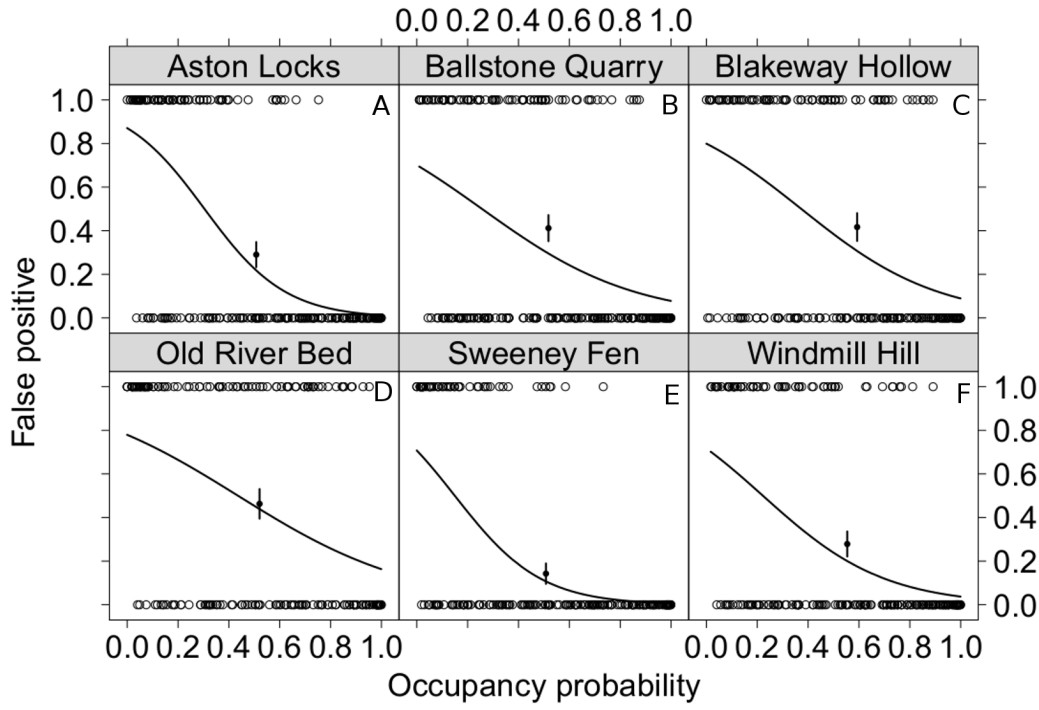

**Figure 3** **The relationship between the commonness of a species and the likelihood of it being observed when it is not there.** False-positive and true-positive observations for each species modelled as a function of the 4 km² occupancy probability of these species in southern England. False-positives were modelled as a binary response variable. To evaluate the goodness of fit of these models the proportion of false-positive errors was calculated from the central third of the data. It is shown with the point on the graphs together with its standard error. All model parameters are highly significant $p < 0.01$. The back transformed coefficients and their 95% confidence intervals where (A) Aston Locks, intercept 0.871 [0.787 –0.928], coefficient $1.89 \times 10^{-3}$ [$3.57 \times 10^{-4} - 7.91 \times 10^{-3}$] d.f. 240; (B) Ballstone Quarry, intercept 0.701 [0.579–0.803], coefficient $3.48 \times 10^{-2}$ [$1.17 \times 10^{-2} - 9.16 \times 10^{-2}$] d.f. 246; (C) Blakeway Hollow, intercept 0.800 [0.690–0.882], coefficient $2.37 \times 10^{-2}$ [$7.72 \times 10^{-3} - 6.45 \times 10^{-2}$] d.f. 227; (D) Old River Bed, intercept 0.780 [0.667–0.866], coefficient $5.23 \times 10^{-2}$ [$1.89 \times 10^{-2} - 1.28 \times 10^{-1}$] d.f. 198; (E) Sweeney Fen, intercept 0.708 [0.575–0.817], coefficient $2.70 \times 10^{-3}$ [$3.62 \times 10^{-4} - 1.43 \times 10^{-2}$] d.f. 215; (F) Windmill Hill, intercept 0.716 [0.586–0.822], coefficient $1.52 \times 10^{-2}$ [$4.02 \times 10^{-3} - 4.88 \times 10^{-2}$] d.f. 229.

## An evaluation of combining the results of the multiple surveys

Increasing the threshold for acceptance of an observation dramatically reduced the number of false-positive observations (Fig. 4). Even when only two surveys are combined the average number of false-positive observations is 8% of the number in a single survey and false-positive observations are practically eliminated if five surveys are combined.

Two useful metrics for test performance are the specificity and sensitivity. Figures 5 and 6 show how the average specificity and sensitivity of plant observation change with different numbers of repeat surveys, using an acceptance threshold of presence in two surveys. With just two surveys the specificity of surveying is close to its maximum value (Fig. 5). However, the increase in specificity has been at the expense of sensitivity, as combining two surveys creates many false-negatives (Fig. 6). Yet with additional surveys the sensitivity rises quickly and exceeds the sensitivity of a single survey, whereas the specificity declines only slowly.

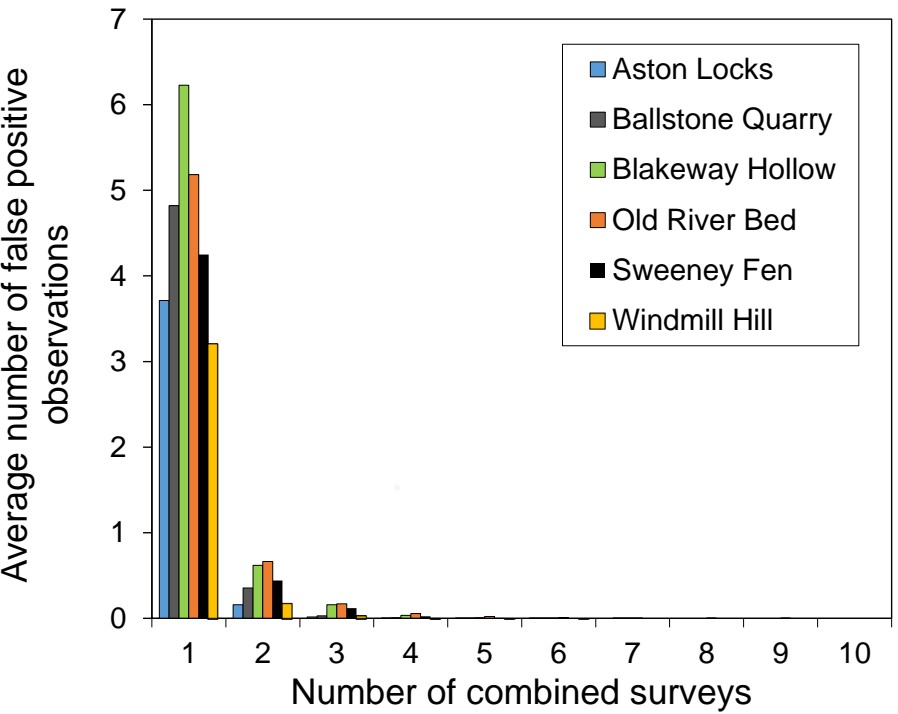

**Figure 4** **Combining the results surveys dramatically reduced the number of false-positive observations.** The reduction in false-positive observations when the threshold for observation acceptance is increased. The graph shows the average number of false-positive observations with either one survey or two or more surveys combined. Apart from when there is one survey, species are only accepted as present at a site if they are present in each survey.

With three surveys the sensitivity is approximately the same as a single survey, but the specificity has improved considerably.

A common strategy for the analysis of botanical records is to combine the results of multiple surveys, so that a species only has to be observed once to be included. The sensitivity of such a strategy increases with increasing numbers of surveys (Fig. 6). However, owing to the accumulation of false-positive observations the specificity is poor and decreases with increasing numbers of surveys (Fig. 5).

## DISCUSSION

There can be few datasets where the same sites have been surveyed independently by such large numbers of observers over such short periods. As the data were collected under exam conditions the independence of the surveys is largely assured, but also there is a clear incentive for observers to minimize their errors. The habitats, methods and participants are comparable to surveys conducted routinely by professional and amateur surveyors. The surveys used in this study were largely conducted on grassland of various types, during the summer. Grassland is used in FISC assessments, because of its species richness and density, but also because it contains a wide variety of species that are known to be difficult to identify. These data are not particularly suitable for analysing habitat and seasonal

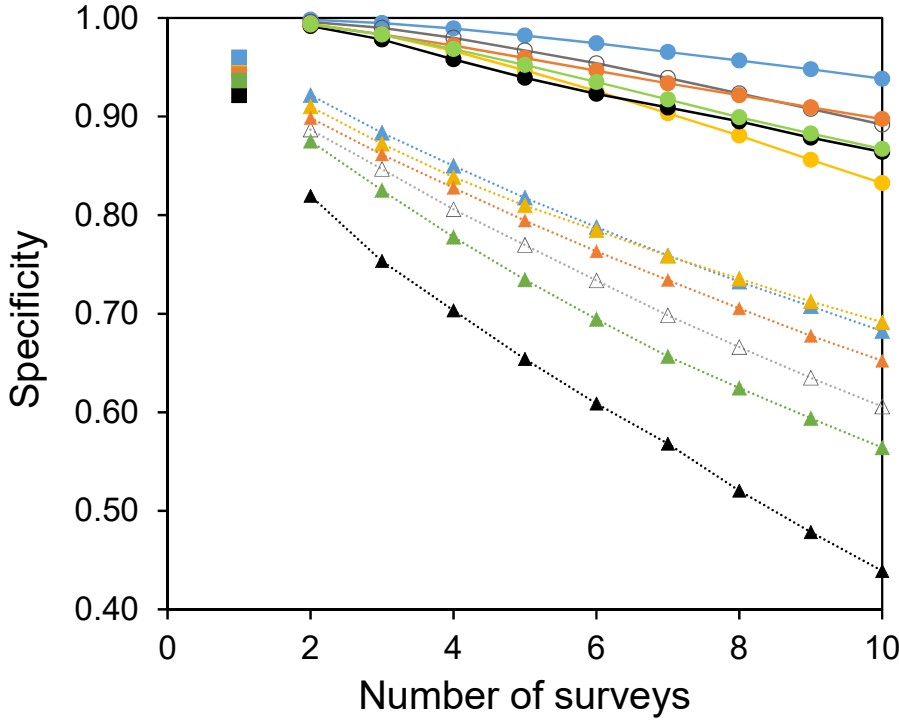

**Figure 5  The specificity of surveys at each site.** Circles with solid lines are where species are only accepted if they were found in two of the multiple surveys. Squares are where there was only one survey. Triangles with dotted lines are where all observed species are accepted. Colours represent the same sites as in Fig. 4. Specificity is calculated from the number of true negatives, divided by the sum of the true negatives and false-positives.

dependent difference in surveying errors, but we believe that the conclusions drawn from these data are also applicable to botanical surveying in general. Moreover, many of the conclusions are likely to be applicable to surveys of other organisms, particular sessile organisms where identification is critical.

A potential criticism of the results is that there may have been errors in the surveys used as a baseline for the results. Nevertheless, we believe the number of these errors will be small and will not have biased the conclusions. We believe this because the results of more expert surveyors converge with the baseline data (Fig. 1). Also, there are no species seen by most of the observers, but not accepted as part of the baseline data. Furthermore, although errors in the baseline data could cause an inflation of the absolute numbers of false-positive observations, our conclusions are largely based on the relative number of false-positives, which would be unaffected by small errors in the baseline.

Botanical surveyors vary considerably, both in their ability to detect species and in their degree of caution (Fig. 1). In the surveys presented here the participants had a clear disincentive to avoid false-positive errors, whereas in many biological surveys there are in fact incentives to create false-positive errors. For example, amateur recording events such as bioblitzes and birding challenges incentivize the creation of long species lists. Having said that, *Farmer, Leonard & Horn (2012)* investigated the influence of incentives on errors in

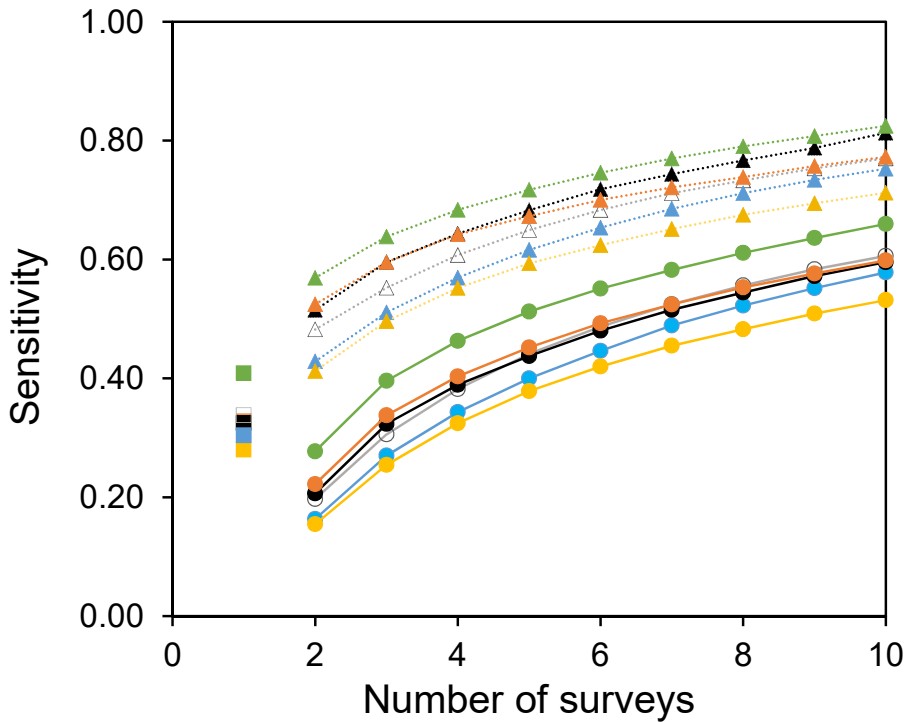

**Figure 6 The sensitivity of surveys at each site.** Circles with solid lines are where species are only accepted if they were found in two of the multiple surveys. Squares are where there was only one survey. Triangles with dotted lines are where all observed species are accepted. Colours represent the same sites as in Fig. 4. Sensitivity is calculated from the number of true positive observations, divided by the sum of the true positive observations and the false-negatives.

bird identification and found no effect. They did, however, see widespread overconfidence among observers, which is consistent with large numbers of false-positive errors created by participants in these surveys.

False-positive errors could be found and corrected for if they occur predictably, *a priori* we expected that we would see a phylogenetic signal in false-positive errors. For example, it is well known that inexperienced botanists find the Poaceae and Cyperaceae difficult to identify. Furthermore, it is not unreasonable to expect recorders to mistake one species for its near relative. Yet overall there is no clear phylogenetic signal to false-positive errors. Therefore, phylogeny does not provide any solution to avoiding false-positive errors and it appears that errors are picked somewhat haphazardly from the pool of potential species. While observers may be less comfortable identifying difficult groups there is little evidence that more false-positive observations are created for these groups. Indeed, the difficulty of some groups could even result in fewer false-positive observations, if their difficulty leads to greater caution by the observers. Nevertheless, the positive correlations of false-positive and cautious observations with the number of species in a genus indicate that errors are more likely for groups with similar identifying features.

Although cautious records are collected in surveys they are rarely used in analyses. It is certainly preferable for observers to make cautious records, rather than making

false-positive mistakes; however, we found no evidence that observers who made more cautious records also recorded fewer false-positive errors.

At all sites, regionally rare species are more likely than common species to be falsely observed. This trend is not restricted to plant identification, the same effect also occurs in bird surveys (*Manel et al., 1999*; *Farmer, Leonard & Horn, 2012*). Knowledge of the regional abundance of species could be a useful tool for the automated validation of observations.

As expected, increasing the threshold for the acceptance of an observation increased the specificity of surveying at the expense of sensitivity. In this paper the acceptance threshold has been varied by requiring multiple observations for the acceptance of a species. The acceptance threshold for an observation will influence the specificity and sensitivity of that survey. However, there is no perfect threshold, and it should be chosen by the experimenter based upon the requirements of the survey. For example, in comparison to using the results from a single survey, surveying a site three times and using an acceptance threshold of presence in two independent surveys will considerably improve the specificity without a loss in survey sensitivity. Nevertheless, this has been at the cost of three times as much surveying effort. In many cases setting such a high threshold for the acceptance of a species at a site is unacceptable, but, there may be occasions when such a threshold is appropriate. Furthermore, it is not necessary to set the same acceptance threshold for all species. Rare species, for example, are more likely to be recorded as false-positive observations, so they might need a higher threshold. Furthermore, although we found no phylogenetic component to false-positive observations, there was considerable variability between species with respect to their likelihood of being a false-positive. If it were known which species are most susceptible to these errors then the threshold for accepting them could be greater.

There are other methods for increasing the threshold for acceptance that can be used by observers in the field. Systems for data entry that give feedback to the user on the likelihood of their observation would give users a chance to reflect on the data they have entered, thus avoiding common slips. Furthermore, IT systems that can support field identification with illustrations and keys, would avoid knowledge-based errors, particularly if such a system was geographically aware. Certainly, at least in Europe, considerable data on land cover, vegetation classification and species distributions exist, which, if it were available to field recorders, could reduce their chances of mistakes.

Were systems for automatic data validation available for field data collection then some predictable errors could be prevented. Errors such as non-existent species are completely preventable, but alerts for locally rare species would also reduce errors. Furthermore, errors due to an incorrect locality or date can be largely eliminated. Errors of common species are still harder to resolve, but simple slips can be validated by using feedback to the observer, particularly if there are already data available from the site for comparison.

Slips and lapses are common types of subconscious errors which, upon review, the observer would know is wrong (*Rasmussen, 1983*). An example of a slip is writing down one species when a similarly named species was intended (e.g., *Carex divisa* instead of *Carex divulsa*) or the transposition of the numbers in coordinates. An example of a lapse is

the omission of important information, such as writing down "*P. vulgaris*" in a situation where "*P.*" might refer to either *Pinguicula*, *Polygala*, *Primula*, *Prunella* or *Pulsatilla*.

Rules-based errors happen when people are working in familiar situations and applying routine rules that they have learned as part of their work, but these rules can either be unsuitable for all occasions, or can be misapplied. In the case of biological recording they may occur when the observer is in an area they do not know well, and applies rules of thumb that work in their familiar area, but not in other areas where a different suite of species are present. Mistakes such as these would be reduced by facilitated access to identification keys in the field, particularly highlighting distinguishing characters.

Knowledge-based rules are where people are confronted with unfamiliar situations and try to apply their knowledge to come to a solution using more in-depth reasoning than simply applying rules. However, they may not appreciate the limits of their own knowledge and are unaware of the potential for error. Such errors come about due to overconfidence and can be compounded by confirmation bias. *Wolf et al. (2013)* show that people working together or at least with the knowledge of other people's decisions will create fewer false-negatives and fewer false-positives. *Senders & Moray (1991)* succinctly stated that "*The less often errors occur, the less likely we are to expect them, and the more we come to believe that they cannot happen*". Thus, biological recording methodologies that fail to inform the observers of their errors are destined to add to the problem by contributing to their complacency.

Designing botanical surveys and analysing their results is particularly difficult due to the large number of species involved, which vary in their rarity and detectability. It is practically impossible to optimise methods for all species simultaneously and even optimising to the average ignores the fact that the species at either extreme of rarity are likely to be the ones we are most interested in. *MacKenzie & Royle (2005)* analysed the efficiency of survey design and concluded that rare species are more efficiently surveyed with extensive surveys, with less intensity at each survey site, whereas efficient surveying for common species was best under intense surveying of fewer sites. Unfortunately, extensive surveying for rare species, without systems to reduce false-positive observations, will create more false-positive observations, reducing the benefits of increased efficiency. Intensive surveying of a few sites for common species is unlikely to be problematic, because common species are rarely false-positives. *MacKenzie & Royle (2005)* also found that there was no efficiency gain in double-sampling, compared to single surveys, though they did not consider the potential advantages of reducing false-positive observations.

In summary, false-positive observations are a common and pervasive problem in botanical surveying. The fact that they occur should not be ignored, and when field survey methods are designed they should be considered. The use of electronic data entry systems in the field should be promoted as their potential to validate, prompt and guide observers will reduce errors.

## ACKNOWLEDGEMENTS

The authors acknowledge the help of Gordon Leel and Mark Duffell, who did the digitization.

### Funding

This work was supported by a grant from the Botanical Society of Britain and Ireland and the EU BON project funded by the EU Framework Programme (FP7/2007-2013) under grant agreement no. 308454. The funders had no role in study design, data collection and analysis, decision to publish, or preparation of the manuscript.

### Grant Disclosures

The following grant information was disclosed by the authors:
Botanical Society of Britain and Ireland.
EU Framework Programme: FP7/2007-2013.

### Competing Interests

The authors declare there are no competing interests.

### Author Contributions

- Quentin J. Groom conceived and designed the experiments, analyzed the data, contributed reagents/materials/analysis tools, wrote the paper, prepared figures and/or tables, reviewed drafts of the paper.
- Sarah J. Whild conceived and designed the experiments, performed the experiments, contributed reagents/materials/analysis tools, wrote the paper, reviewed drafts of the paper, coordinated digitization of the data.

### Data Availability

Zenodo, 10.5281/zenodo.46662, http://doi.org/10.5281/zenodo.46662.

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
