# Peer review of "Characterisation of false-positive observations in botanical surveys"

_PeerJ, doi:10.7717/peerj.3324_

## Round 0.1 · original submission · Major Revisions

The comments from the three reviewers comprise a mixed bag, ranging from minor revision to reject. However, I did note that the reject decision (Reviewer 2) did include an option for re submission subject to addressing a substantive concern. And that concern is: what is the error associated with the "gold standard". I agree with the reviewer that this needs to be addressed, and doing so will be difficult. Reviewer 3's comments are, in my opinion, easier to deal with involving as they do issues of clarification.

·

Basic reporting

This paper is generally well-written and the study is placed in the appropriate context. The tables and figures are well done and appropriate.

Experimental design

The data set evaluated is large and appears to be an appropriate one for this type of study, and I agree with the authors that it is one of only a few of its kind. A number of studies have documented pseudoturnover in species lists among multiple observers, but false-positive errors have rarely been evaluated, presumably due to the lack of a gold standard for comparison.

Validity of the findings

The statistics seem appropriate, and the conclusions are generally well supported by the data.

Additional comments

I do not have any major concerns but a number of minor ones that are annotated on the PDF.

Reviewer 2 ·

Basic reporting

This paper addresses an important question - the degree to which false positive and false negative errors occur in botanical surveys. The paper is quite well presented. However, the paper has one major flaw in that false positives can only be assumed to occur if the "gold standard" of presences and absences is in fact flawless. The authors assumed that for their analyses, but acknowledged that the assumption might be incorrect. In this paper, the "gold standard" was obtained from identifications of a highly-experienced surveyor. However, the degree to which this gold-standard might include false negatives (or false positives) does not seem to have been quantified. The consequence is that the degree of errors in the gold standard are unclear.

At this point, it is difficult for the authors to rectify this issue. Ideally a second experienced surveyor (or more) would have undertaken the same surveys. Further, the experienced surveyors would have been permitted to search the sites for a time period that was longer than the other surveyors. Without knowing the false negative and false positive rates in the gold standard, the analyses are unable to accurately estimate these rates for the other surveyors.

Minor comments

In the abstract the meaning of "threshold for the acceptance of an observation" is unclear. This threshold has not been described previously in the abstract.

"False-positive errors are higher in field surveying of plants than many people may appreciate." This might be true, but this study only shows that false positives are relatively common in this one case. Other instances might have different false-positive rates, particularly if the incentives to balance false-positive and false-negative errors were different.

"Combining multiple surveys" What are these multiple survey - by different people, or separate surveys by the same person(s)? How should they be combined?

Would taking specimens and seeking input from others or keys/guides help? This is discussed somewhat, but the degree to which these actions might help is unclear.

It is debatable that errors are "more often than not" the result of human fallibility. This claim is not supported by data or a citation. It seems to me that random errors will occur, to some degree, in almost all empirical studies. Human errors will occur in many fewer studies, I would hope. Therefore, I think this sentence needs tempering. How about "Sometimes they result from random chance, but also often from human fallibility." That way there is no claim about the relative extent of the sources of the errors (assuming no such data can be cited as evidence).

The paper by Ellis (1991), cited as evidence that "much of expert plant identification in the field is done using gestalt perception", is a study of quite limited scope - personal observations by the author of some particular case studies by potentially amateur or inexperienced naturalists. Again, this statement might by true, but the paper by Ellis is not sufficient evidence to state categorically that it is true.

Line 97: "... for a small proportion ... "

Experimental design

The scope and general approach is good, with the exception that the fallibility of the gold-standard remains unknown (see above).

Validity of the findings

See above.

Reviewer 3 ·

Basic reporting

The manuscript is for the most part well-written – the language is both scientific and accessible, which assists the reader greatly. The Introduction begins well, with the authors setting out the problem nicely, but does not live up to its initial promise. Sections of the Introduction miss opportunities to draw on an existing body of literature. In particular, the section on the relative value of false negatives and false positives could draw on the sizable literature in this space (for example, Royle & Link (2006) Ecology, 87:835-841; Wintle et al (2012) Diversity & Distributions, 18:417-424; Garrard et al (2015) Conservation Biology, 29:216-225). In addition, there would be a number of references within the species distribution modelling literature that would help to back up claims about the costliness of false positive observations.

In general, I believe this manuscript suffers from some structural problems that contribute to a lack of clarity about what exactly was done, how it is was done and how the results can/should be interpreted. A number of concepts and terms are not defined or explained at their first usage. For example, words like apophenia (line 59), I think warrant a brief definition, and phylogenetic signal tests are introduced in line 166, without reference to what they are and how they are relevant to the key goals of the study. In addition, some information reported in the results (eg. line 253: “If surveyors were not able to identify a plant to species, they could alternatively report a genus”; and lines 280-284 describing specificity and sensitivity), seem to be much better placed in the Methods (and would help the reader to better understand the Methods enormously).

I also found that some of the terminology lacked clarity (for example, is the “false detection probability” referred to in line 223 actually the probability of a false positive observation occurring?). And, for example, in the discussion about specificity and sensitivity, it would be useful (given that the paper is about false positives and, to a lesser extent, false negatives) for the authors to acknowledge that these are equivalent to the probability of true negatives and true positives, which might assist the reader.

Experimental design

The authors are correct when they state that they have access to a unique dataset (lines 298-299) – it is truly a valuable dataset. However, I believe that the methods section of the manuscript does not meet expectations. Although the authors have supplied the data and codes, I do not believe that the methods could be reasonably replicated based on what is presented. This is partly because the research questions are poorly articulated (if at all). For example, the reader is introduced to the phylogenetic signal tests in the Data section, without any description of what these tests are (this comes later in lines 191-199) or, more importantly, WHY they were performed.

I found the description of the statistical tests particularly confusing. For example, it was not clear to me what independent variables were considered or modelled in the GLMs described in lines 185 – 190. I think regional frequency was one, but even this is very clear. Perhaps it might help in this section to provide more information, and perhaps a linear equation highlighting how the response variable is thought to be related to any potential independent variables?

For this manuscript to be publishable, I believe the Methods section would require a thorough edit to better explain exactly what analyses were completed and how they link back to research questions and objectives.

Validity of the findings

Given my comments above, it is difficult for me to comment on the validity of the findings, as it is not clear to me exactly how statistical analyses were done, and whether they have sufficient statistical rigour. As such, I have found it difficult to comment thoroughly on the results and discussion sections of this manuscript.

I agree with the authors that the problem of false positives is paid much less attention than the problem of false negatives in botanical surveys, but I do not think the authors have adequately established that false positives SHOULD be paid more attention (although I think they could do so, as mentioned above, by highlighting the impact that false positives have on commonly employed techniques like species distribution modelling, as well as on environmental management, monitoring and protection policy). Nor have they demonstrated clearly how their research will address the problem.

Additional comments

I provide the following specific comments, which may help to clarify some of my concerns as outlined in Sections 1-3 above.

Lines 68-69: I think this statement requires more detailed discussion, justification and referencing! Eg. more costly in terms of what? This is a very complex trade-off and will depend entirely on a person’s perspective about what is important. Also, in framing the problem in this way, the interactive nature of false positives and false negatives is played down. For example, when considering a single survey where the aim is to build a complete list of the species at a site (but not interested in abundance or cover), then a false positive observation for one species will often create a false negative observation for another.

Lines 78-80: I do not think this statement is evidence of the costs associated with false positives. I’m sure there are better ways to justify investment in minimizing false positive errors (including the valid reasons you have mentioned in lines 73-76.

Lines 122-124: Why were these types of habitat selected for? Is this to do with access/ease of movement, or related to the number and type of species as you imply later?

Lines 149-151: To what extent do you think that providing such a disincentive is reasonable? Did candidates also lose marks for false negatives? Has this disincentive simply promoted overly cautious responses?

164- 165: can you please provide more information about the different types of species, and how the team identifies each category?

216: How was detection probability calculated?

276: I cannot find an explanation in the methods of the process used to increase the threshold for acceptance, making this result extremely difficult to interpret.

---

## Round 0.2 · accepted · Accept

I believe your paper has made a substantive contribution to a poorly documented problem. As you state in the paper, this is definitely not the last word on this topic. Overall, you have handled the reviewer's comments well. While you may not have achieved the ultimate "gold standard", the paper has many useful pointers for further research.